# Robust Direction of Arrival and Polarization Parameter Estimation in Mutual Coupling Scenario with Non-Collocated Crossed Dipole Arrays

**DOI:** 10.3390/s25051391

**Published:** 2025-02-25

**Authors:** Wenjiang Chen, Xiang Lan, Xianpeng Wang

**Affiliations:** School of Information and Communication Engineering, Hainan University, Haikou 570228, China; 22220854000095@hainanu.edu.cn (W.C.); wxpeng2016@hainanu.edu.cn (X.W.)

**Keywords:** distributed crossed dipole array, mutual coupling effects, subarray selection

## Abstract

Traditional direction of arrival (DOA) and polarization parameter estimation algorithms generally perform well under ideal conditions. However, their performance degrades significantly in practical scenarios due to mutual coupling effects among array elements. This work introduces an innovative method based on a distributed crossed dipole array to jointly estimate DOA and polarization parameters in the presence of mutual coupling effects. This work firstly eliminates the mutual coupling matrix (MCM) through subarray selection, without requiring prior knowledge of the array’s mutual coupling. The DOA is then estimated using an improved high-resolution algorithm, followed by accurate estimation of the polarization parameters through parameter matching. The results from simulations confirm that the new method significantly improves the estimation accuracy in complex mutual coupling environments, showing notable potential for practical applications and robust performance.

## 1. Introduction

The increasing complexity of modern signal processing environments has made precise target localization and parameter estimation critical in various applications, including radar systems, wireless communications, and electronic warfare. However, these environments often involve multiple targets and strong interference signals, posing significant challenges to traditional array signal processing techniques [1,2]. Current array signal processing methods mainly rely on spatial information collected by array antennas. While classical high-resolution parameter estimation algorithms, such as the MUSIC (Multiple Signal Classification) algorithm and the ESPRIT (Estimation of Signal Parameters via Rotational Invariance Techniques) algorithm, enable accurate target localization and parameter estimation [3,4], these methods typically depend on scalar sensors. To improve computational efficiency and enable real-time DOA estimation, phase interferometry-based methods and full-hardware in-phase/quadrature (I/Q)-based angle of arrival (AoA) estimation techniques have been proposed [5]. Phase interferometry exploits phase differences among received signals to estimate the AoA while reducing computational complexity, making it suitable for real-time applications. This technique provides a compelling alternative to traditional subspace-based algorithms by achieving a trade-off between computational efficiency and estimation accuracy. However, scalar sensors capture only the signal’s spatial characteristics, thereby neglecting its polarization characteristics. This limitation becomes especially critical in modern complex environments, such as multi-target scenarios and strong interference, where methods relying solely on spatial data struggle to effectively distinguish target signals from interference. In order to overcome these limitations, the polarization-sensitive array (PSA) has evolved from the traditional scalar array to spatial polarization information fusion by introducing electromagnetic vector sensors (EMVSs). Unlike scalar arrays, PSAs can simultaneously capture both the directional and polarization characteristics of signals, thus offering a multidimensional perspective in signal analysis. By exploiting polarization differences between target and interference signals, the PSA facilitates more accurate signal separation and interference suppression. Consequently, the PSA proves especially effective in complex environments involving multiple sources and significant interference. This joint processing of spatial and polarization information significantly improves the system’s robustness in complex signal environments [6,7,8,9,10].

In response to the limitations of scalar arrays, research on PSAs has recently focused on the joint DOA and polarization parameter estimation technique, which significantly improves the array’s signal processing capability by fusing spatial and polarization information. Zhang investigated a blind DOA and polarization parameter joint estimation algorithm based on dimensionality-reducing MUSIC [11]. This method improves the estimation accuracy under irregular arrays by reducing the complexity of multidimensional search and performs well in complex environments. However, its performance degrades significantly under mutual coupling effects, as it does not include explicit coupling mitigation. In [12], Shuai proposed a quaternion MUSIC algorithm based on vector MISC (maximum inter-element spacing constraint) arrays, which further optimizes the precision and computational performance in estimating multidimensional parameters. This algorithm requires a large array aperture, increasing hardware complexity. To achieve comprehensive estimation of four-dimensional (4-D) parameters, Lan proposed an innovative approach by replacing traditional crossed dipoles with tripoles to construct a linear tripole array. This array exhibits high degrees of freedom, enabling the accurate acquisition of two-dimensional (2-D) DOA and polarization information [13]. Furthermore, Lan developed an optimized MUSIC approach based on linear triplet arrays to overcome the challenge of intensive resource usage in high-dimensional computations. By decomposing the 4-D DOA and polarization estimation into two independent 2-D analyses, this method notably reduces computational effort while enhancing signal resolution [14]. Nevertheless, the tripole array inherently suffers from severe mutual coupling effects, as three closely positioned orthogonal dipoles within each sensor element introduce strong internal interactions, which can significantly degrade estimation accuracy in practical applications. This internal coupling necessitates complex calibration techniques and increases computational burden, particularly in high-source-density scenarios. In addition, Dai et al. proposed an innovative method by introducing element rotation in linear dipole arrays (LDAs), significantly enhancing their 2-D DOA and polarization estimation capabilities [15]. By optimizing element orientations, this method improves estimation accuracy and resolution. But, the need for the precise mechanical control of element rotation increases complexity, posing challenges for practical implementation. Meanwhile, a new method based on a single polarization vector sensor was proposed in [16], which simplifies the hardware design without relying on large arrays while achieving highly accurate DOA and polarization parameter estimation. The sensitivity of this method to noise and mutual coupling effects reduces its robustness in practical applications.

Although these parameter estimation algorithms mentioned above perform well in simulations and controlled environments, in practical works, their performance is affected by various non-ideal factors, such as amplitude and phase errors, non-ideal noise, and mutual coupling effects. As EMVSs in polarization-sensitive arrays usually consist of multiple dipoles co-located at the same spatial coordinates, they inevitably experience mutual coupling during signal reception. Such coupling arises not only among the dipoles themselves but also between neighboring array elements [17]. This distortion critically impacts the estimation accuracy of DOA and polarization parameters, leading to significant performance degradation in practical applications [18]. Mutual coupling effects cannot be eliminated, and although complex electromagnetic isolation in hardware can significantly reduce mutual coupling, this will make the hardware cost increase [19]. In order to address the effect of the mutual coupling effect on the DOA estimation performance, some researchers have adopted special antenna arrangements: Wong in the reference [20] separated six closely arranged antennas of EMVSs into spatially non-collocated antennas, thereby enabling precise geometric layout and vector processing to capture the target signals’ spatial information. One approach is based on subarray selection and mutual coupling matrix estimation, which can improve the accuracy of DOA estimation to some extent [21]. Recent studies have explored linear techniques for artifacts correction and compensation in AoA estimation. Florio et al. [22] introduced an α-matrix-based linear compensation approach that embeds systematic error corrections and first-order mutual coupling compensation into a precomputed correction matrix. This approach is computationally efficient and has been experimentally validated to significantly reduce estimation errors in phase interferometric AoA estimation. Furthermore, reference [23] proposed a parallel synthetic coprime polarization-sensitive array (PSC-PSA) that leverages a non-collocated design and compressed sensing for joint DOA and polarization estimation. This approach effectively enhances estimation accuracy by expanding the array aperture and reducing mutual coupling effects, making it suitable for complex signal environments. Although existing methods have made significant progress in mutual coupling correction, they require high computational resources; therefore, accurately estimating DOA and polarization parameters while maintaining simplified hardware remains a critical research challenge. This challenge becomes more complex when mutual coupling effects are considered. To address these challenges, this paper proposes an innovative method for DOA and polarization parameter estimation under the influence of mutual coupling effects. A new non-collocated polarization-sensitive array model is utilized to mitigate mutual coupling interference [24,25]. In addition, an improved-ESPRIT algorithm is developed to achieve decoupling by combining subarray selection methods to eliminate the effect of the MCM on parameter estimation. The main contributions of this paper are summarized as follows:(1)The non-collocated polarization array model is further refined. Although prior works have attempted to apply similar models, the focus here is on mitigating mutual coupling between dipoles within the EMVS (i.e., inter-polarization coupling, IPC) and reducing coupling among neighboring array elements (i.e., inter-elemental coupling, IEC) via a uniform linear array configuration. Moreover, the proposed model greatly enlarges the array’s effective aperture, lowers hardware expenses, and achieves a more stable parameter estimation in complex mutual coupling environments.(2)We propose an improved ESPRIT algorithm. The proposed algorithm first decouples the MCM using a novel subarray selection strategy and subsequently performs parameter matching, effectively eliminating the impact of mutual coupling on parameter estimation. In addition, the algorithm does not need to rely on the a priori knowledge of MCM, which makes it more robust and easy to implement in practical applications. Compared with the traditional algorithms, the improvement significantly enhances the accuracy and stability of parameter estimation under strong mutual coupling conditions.(3)The proposed algorithm notably reduces the overall computational complexity through a non-collocated array design and simplified subarray selection strategy. Unlike traditional algorithms that rely on complex EMVS designs with multiple co-located dipoles and high-dimensional optimization, the proposed method reduces mutual coupling interference and achieves high-precision joint parameter estimation within a low-complexity framework.

This paper is structured as follows. The signal model of the non-collocated polarized array and the effects of mutual coupling on the array dipoles, including the mutual coupling matrix, are presented in Section 2. The detailed procedure of the proposed algorithm is presented in Section 3. In Section 4, the experimental simulation results and discussion of the algorithm are provided. Section 5 presents the conclusion of this work.

## 2. Signal Model

### 2.1. Data Model

As shown in Figure 1, a uniform linear array (ULA) consists of *M* groups of separated dipoles. For convenience of description, the dipoles are referred to as ’elements’ in this paper. All array elements are uniformly spaced along the *y*-axis. Each group consists of two orthogonal components, aligned parallel to the *x*-axis and *y*-axis, respectively. The spacing *d* between two adjacent element groups is set to $\frac{\lambda }{2}$, while the distance between the two orthogonal elements in each group is $\frac{\lambda }{4}$, where λ is the wavelength of the signal.

Consider that *K* far-field uncorrelated narrowband electromagnetic wave signals impinge on the polarization-sensitive array. The polarization-sensitive array carries out joint sampling in the space–polarization domain; $\theta \in \left[-\pi /2,\pi /2\right)$ and $\varphi \in \left[-\pi ,\pi \right)$ denote the elevation angle and azimuth angle of the incident signals, respectively; $\gamma \in \left[0,\pi /2\right)$ and $\eta \in \left[-\pi ,\pi \right)$ denote the incident signals’ auxiliary polarization angle and the polarization phase difference. The space–polarization domain information of the *k*th received target can be denoted as $a_{k}\left(\theta_{k},\varphi_{k},\gamma_{k},\eta_{k}\right)$, which comprises the space vector $a_{s_{k}}\left(\theta_{k},\varphi_{k}\right)$ and the polarization vector $a_{p_{k}}\left(\theta_{k},\varphi_{k},\gamma_{k},\eta_{k}\right)$, defined as follows:(1)$$a_{s_{k}}\left(\theta_{k},\varphi_{k}\right)=\left[\begin{array}{c} 1\\ e^{-j2\pi \frac{d}{\lambda }\sin\theta_{k}\sin\varphi_{k}}\\ \vdots \\ e^{-j2\pi \left(M-1\right)\frac{d}{\lambda }\sin\theta_{k}\sin\varphi_{k}} \end{array}\right]$$(2)$$a_{p_{k}}\left(\theta_{k},\phi_{k},\gamma_{k},\eta_{k}\right)=\left[\begin{array}{cc} \cos\theta_{k}\cos\varphi_{k} & -\sin\varphi_{k}\\ q_{k}\cos\theta_{k}\sin\varphi_{k} & q_{k}\cos\varphi_{k} \end{array}\right]\left[\begin{array}{c} \sin\gamma_{k}e^{j\eta_{k}}\\ \cos\gamma_{k} \end{array}\right]$$
where $q_{k}=e^{-j2\pi \frac{d}{\lambda }\sin\theta_{k}}$ denotes the spatial phase factor between two elements in a set of element groups for the *k*th target. Let azimuth angle φ of each incident signal be $\frac{\pi }{2}$ in this paper; Equations (Equation 1) and (Equation 2) can be simplified as(3)$$a_{s_{k}}\left(\theta_{k}\right)=\left[\begin{array}{c} 1\\ e^{-j2\pi \frac{d}{\lambda }\sin\theta_{k}}\\ \vdots \\ e^{-j2\pi \left(M-1\right)\frac{d}{\lambda }\sin\theta_{k}} \end{array}\right]$$(4)$$a_{p_{k}}\left(\theta_{k},\gamma_{k},\eta_{k}\right)=\left[\begin{array}{cc} 0 & -1\\ q_{k}\cos\theta_{k} & 0 \end{array}\right]\left[\begin{array}{c} \sin\gamma_{k}e^{j\eta_{k}}\\ \cos\gamma_{k} \end{array}\right]$$

The *k*th joint steering vector $a_{k}\left(\theta_{k},\gamma_{k},\eta_{k}\right)$ of the PSA is denoted as(5)$$\begin{array}{cc} a_{k}\left(\theta_{k},\gamma_{k},\eta_{k}\right) & =a_{s_{k}}\left(\theta_{k}\right)⊗a_{p_{k}}\left(\theta_{k},\gamma_{k},\eta_{k}\right)\\ \; & =\left[\begin{array}{c} 1\\ e^{-j\frac{\pi }{2}\sin\theta_{k}}\\ \vdots \\ e^{-j\left(M-1\right)\frac{\pi }{2}\sin\theta_{k}} \end{array}\right]⊗\left[\begin{array}{cc} 0 & -1\\ q_{k}\cos\theta_{k} & 0 \end{array}\right]\left[\begin{array}{c} \sin\gamma_{k}e^{j\eta_{k}}\\ \cos\gamma_{k} \end{array}\right] \end{array}$$
where ⊗ denotes the Kronecker product, and the signal received at the output of this array is(6)$$X\left(t\right)=\left[a_{s_{1}}⊗a_{p_{1}},a_{s_{2}}⊗a_{p_{2}},\ldots ,a_{s_{K}}⊗a_{p_{K}}\right]\left[\begin{array}{c} s_{1}\left(t\right)\\ s_{2}\left(t\right)\\ \vdots \\ s_{K}\left(t\right) \end{array}\right]+N\left(t\right)$$
where $s_{k}\left(t\right)$ denotes the *k*th signal received, and N(t) is a 2M×1 Gaussian white noise vector. Then, the signal X(t) can be expressed as(7)$$\begin{array}{cc} X(t) & =\left(A_{s}⊙A_{p}\right)S\left(t\right)+N\left(t\right)\\ \; & =\mathrm{AS}(t)+N(t) \end{array}$$
where $A_{s}=\left[a_{s_{1}},a_{s_{2}},\ldots ,\right.$$\left.a_{s_{K}}\right]$ is the M×K dimensional steering matrix, $A_{p}=\left[a_{p_{1}},a_{p_{2}},\ldots ,a_{p_{K}}\right]$ is the 2×K dimensional polarization matrix, $A=[a_{1},a_{2},\ldots ,a_{K}]$ is the 2M×K-dimensional joint space–polarization steering matrix, and ⊙ denotes the Khatri–Rao product.

### 2.2. Mutual Coupling

In practical works, mutual coupling between antennas is an essential issue in antenna array design and array signal processing, particularly in PSAs. Antenna mutual coupling refers to the effect of one antenna array element on the neighboring antenna array elements when transmitting or receiving signals. Mutual coupling is mainly caused by electromagnetic field interactions, which are especially obvious in array antennas and unavoidable in practical applications.

Assuming that the elements parallel to the *x*-axis and the *y*-axis are entirely orthogonal, the mutual coupling can be ignored among the orthogonal elements, and only the coupling among those elements parallel to each other needs to be considered [18]. It is assumed that the mutual coupling coefficients between two elements are equal if they are equally spaced, regardless of whether they are aligned along the *x*-axis or the *y*-axis. Moreover, the magnitude of the mutual coupling coefficient is affected by the distance between the two elements. When the spacing exceeds a threshold distance, the mutual coupling effect becomes very small or even negligible. The effect of element coupling influence is shown in Figure 2 and Figure 3.

The mutual coupling matrix effect on a ULA can be described as a banded symmetric Toeplitz matrix C, where(8)$$C={\left[\begin{array}{ccccccccc} c_{1} & 0 & c_{2} & 0 & \cdots & c_{L} & 0 & \cdots & 0\\ 0 & c_{1} & 0 & c_{2} & \cdots & 0 & c_{L} & \cdots & 0\\ \vdots & \vdots & \vdots & \vdots & \vdots & \vdots & \vdots & \vdots & \vdots \\ c_{L} & 0 & c_{L-1} & 0 & \cdots & c_{1} & 0 & \cdots & 0\\ 0 & c_{L} & 0 & c_{L-1} & \cdots & 0 & c_{1} & \cdots & 0\\ \vdots & \vdots & \vdots & \vdots & \vdots & \vdots & \vdots & \vdots & \vdots \\ 0 & 0 & 0 & 0 & \cdots & 0 & c_{L} & \cdots & c_{1} \end{array}\right]}_{2M\times 2M}$$

$c_{l}\left(l=1,2,\cdots ,L\right)$ denotes the mutual coupling coefficient between an element and the *l*th parallel neighboring element, where $c_{1}$ denotes the mutual coupling coefficient of the element to itself, usually $c_{1}=1$. *L* denotes the degree of freedom (DOF) of the MCM, and the final model of the received signal, incorporating mutual coupling interference, is derived from Equation (Equation 7):(9)$$\tilde{X}\left(t\right)=\mathrm{CAS}\left(t\right)+N\left(t\right)$$

The covariance matrix R of the array output is denoted as(10)$$\begin{array}{cc} R & =E\left\{\tilde{X}\left(t\right){\tilde{X}}^{H}\left(t\right)\right\}\\ & =\mathrm{CA}R_{s}A^{H}C^{H}+\sigma_{n}^{2}I_{2M} \end{array}$$
where $R_{s}$ is the autocorrelation matrix of the incident signals S(t). $\sigma_{n}^{2}$ and $I_{2M}$ represent the noise power and 2M×2M identity matrix, respectively. The eigenvalue decomposition of the signal covariance matrix R shown in (Equation 10) yields(11)$$R=E_{s}\Lambda_{s}E_{s}^{H}+E_{n}\Lambda_{n}E_{n}^{H}$$
where $\Lambda_{s}\in C^{K\times K}$ and $\Lambda_{n}\in C^{\left(2M-K)\times (2M-K\right)}$ are the diagonal matrices containing the eigenvalues of the covariance matrix R, respectively. The matrices $E_{s}\in C^{2M\times K}$ and $E_{n}\in C^{\left(2M-K)\times (2M-K\right)}$ correspond to the signal and noise subspace, separately.

In practical work, considering a finite number of snapshots *N*, the covariance matrix $\hat{R}$ is always estimated by the mean value of snapshot multiplication:(12)$$\hat{R}=\frac{1}{N}\sum_{n=1}^{N}\tilde{X}\left(t_{n}\right){\tilde{X}}^{H}\left(t_{n}\right)$$

## 3. Joint DOA and Polarization Estimation

### 3.1. DOA Estimation

The proposed method accomplishes DOA estimation through an improved ESPRIT method combined with careful subarray selection to decouple mutual coupling. A straightforward approach would be selecting the first M−1 groups of elements as one subarray and the last M−1 groups of elements as another; the subarray selection is shown in Figure 4.

The obtained mutual coupling matrices $C_{1}$ and $C_{2}$ are denoted as(13)$$C_{1}={\left[\begin{array}{ccccccccc} c_{1} & 0 & c_{2} & 0 & \cdots & c_{L} & 0 & \cdots & 0\\ 0 & c_{1} & 0 & c_{2} & \cdots & 0 & c_{L} & \cdots & 0\\ \vdots & \vdots & \vdots & \vdots & \ddots & \vdots & \vdots & \ddots & \vdots \\ c_{L} & 0 & c_{L-1} & 0 & \cdots & c_{1} & 0 & \cdots & 0\\ 0 & c_{L} & 0 & c_{L-1} & \cdots & 0 & c_{1} & \cdots & 0\\ \vdots & \vdots & \vdots & \vdots & \ddots & \vdots & \vdots & \ddots & \vdots \\ 0 & 0 & 0 & 0 & \cdots & 0 & c_{L-1} & \cdots & c_{2} \end{array}\right]}_{2(M-1)\times 2M}$$(14)$$C_{2}={\left[\begin{array}{ccccccccc} c_{2} & 0 & c_{1} & 0 & \cdots & c_{L} & 0 & \cdots & 0\\ 0 & c_{2} & 0 & c_{1} & \cdots & 0 & c_{L} & \cdots & 0\\ \vdots & \vdots & \vdots & \vdots & \ddots & \vdots & \vdots & \ddots & \vdots \\ c_{L} & 0 & c_{L-1} & 0 & \cdots & c_{1} & 0 & \cdots & 0\\ 0 & c_{L} & 0 & c_{L-1} & \cdots & 0 & c_{1} & \cdots & 0\\ \vdots & \vdots & \vdots & \vdots & \ddots & \vdots & \vdots & \ddots & \vdots \\ 0 & 0 & 0 & 0 & \cdots & 0 & c_{L} & \cdots & c_{1} \end{array}\right]}_{2(M-1)\times 2M}$$

From Equations (Equation 13) and (Equation 14), it is clear that $C_{1}\neq C_{2}$, which makes it impossible to eliminate the mutual coupling matrix and causes failures to most current algorithms. To address this issue, two new subarrays should be selected to ensure that $C_{1}=C_{2}$. Specifically, the 2L−1th to 2(M−L)th groups are selected to form subarray 1, and the elements from the 2L+1th to 2(M−L+1)th groups are selected to form subarray 2. The selection of subarrays is illustrated in Figure 5.

The mutual coupling matrices $C_{1}$ and $C_{2}$ obtained from the selected new subarrays, $C_{1}$ and $C_{2}$, are taken as the first 2(M−1) columns from row 2L−1 to row 2(M−L) and the last 2(M−1) columns from row 2L+1 to row 2(M−L+1) of the mutual coupling matrix C, respectively, where the obtained $C_{1}$ and $C_{2}$ are denoted as(15)$$C_{1}=C_{2}={\left[\begin{array}{cccccccccc} c_{L} & 0 & \cdots & c_{1} & 0 & \cdots & 0 & c_{L} & \cdots & 0\\ 0 & c_{L} & \cdots & 0 & c_{1} & \cdots & c_{L-1} & 0 & \cdots & 0\\ \vdots & \vdots & \vdots & \vdots & \vdots & \vdots & \vdots & \vdots & \vdots & \vdots \\ 0 & 0 & \cdots & 0 & c_{L} & \cdots & 0 & c_{1} & \cdots & c_{L} \end{array}\right]}_{2(M-2L-1)\times 2(M-L-1)}$$

From Equation (Equation 15), notice that the mutual coupling matrix $C_{1}=C_{2}$ and the corresponding joint space–polarization steering matrices of subarray 1 and subarray 2 are $A_{1}$ and $A_{2}$, which correspond to the first 2M−1 columns and the last 2M−1 columns of the joint space–polarization steering matrix A, respectively. The signal subspaces of subarray 1 and subarray 2 are $E_{s1}$ and $E_{s2}$, which correspond to the first 2M−1 rows and the last 2M−1 rows of the signal subspace $E_{s}$, respectively, and there are(16)$$E_{s1}=C_{1}A_{1}T$$(17)$$E_{s2}=C_{2}A_{2}T=C_{1}A_{1}\Phi T$$
where T is a full-rank K×K-dimensional transformation matrix, and, since the *K* signals are independent of each other, T is a full-rank matrix. Φ is a K×K diagonal matrix; the elements on the Φ diagonal are the *K* signals’ phase differences between the two subarrays, and Φ is called the rotation factor. The shift-invariant property of the two subarrays ensures that their received signals maintain rotational invariance. Consequently, the received signals of subarray 2 are obtained by multiplying the received signals of subarray 1 by a rotational factor Φ, i.e., $A_{1}\Phi =A_{2}$, where Φ is denoted as(18)$$\Phi =diag\{e^{-j\pi \sin\theta_{1}},e^{-j\pi \sin\theta_{2}},\cdots ,e^{-j\pi \sin\theta_{K}}\}$$

From Equation (Equation 18), the elevation angles of the signals can be estimated as long as the rotation factor matrix Φ can be estimated. The mutual coupling matrices $C_{1}$ and $C_{2}$ can be eliminated by associating Equation (Equation 16) with Equation (Equation 17) to obtain(19)$$E_{s2}=E_{s1}T^{-1}\Phi T$$

According to the least squares criterion, we obtain(20)$$E_{s1}^{+}E_{s2}=T^{-1}\Phi T$$

Let $\Psi =E_{s1}^{+}E_{s2}$, which gives(21)$$\Psi =T^{-1}\Phi T$$

Ψ and Φ are mutually similar matrices with the same eigenvalues. Therefore, the eigenvalue decomposition of Ψ can estimate the diagonal elements of matrix Φ, i.e., we can obtain $e^{-j\pi \sin\theta_{k}},k=1,2,\ldots ,K$. We can obtain the estimations of the elevation angle ${\hat{\theta }}_{k}$ by the following equation:(22)$${\hat{\theta }}_{k}=\arcsin\left(-\frac{angle[{\hat{\beta }}_{k}]}{\pi }\right)$$
where ${\hat{\beta }}_{k}$ denotes the *k*th eigenvalue of Ψ, and angle[·] denotes taking the phase angle of the complex number.

### 3.2. Polarization Parameter Estimation

Estimating the polarization parameters can also be performed the same way as estimating the DOA, but two new subarrays must be re-selected. Since the elements parallel to the *x*-axis and parallel to the *y*-axis receive different polarization component information, the polarization difference between the two subarrays is used to estimate the polarization parameters.

The subarrays are selected as shown in Figure 6: subarray 3 and subarray 4 are selected parallel to the *x*-axis and the *y*-axis, respectively. And, their mutual coupling matrices $C_{3}$ and $C_{4}$ are shown in the following Equation (Equation 23):(23)$$C_{3}=C_{4}={\left[\begin{array}{ccccccc} c_{1} & c_{2} & \cdots & c_{L} & 0 & \cdots & 0\\ c_{2} & c_{1} & \cdots & c_{L-1} & c_{L} & \cdots & 0\\ \vdots & \vdots & \vdots & \vdots & \vdots & \vdots & \vdots \\ c_{L} & c_{L-1} & \cdots & c_{1} & c_{2} & \cdots & 0\\ \vdots & \vdots & \vdots & \vdots & \vdots & \vdots & \vdots \\ 0 & 0 & \cdots & 0 & c_{L} & \cdots & c_{1} \end{array}\right]}_{M\times M}$$

From the above Equation (Equation 23), we notice that $C_{3}=C_{4}$, we can similarly estimate the polarization parameters by the ESPRIT method, and the joint space–polarization steering vectors $a_{x_{k}}$ and $a_{y_{k}}$ for subarray 3 and subarray 4 are denoted, respectively, as(24)$$a_{x_{k}}=\left[\begin{array}{c} -\cos\gamma_{k}\\ -e^{-j\frac{\pi }{2}\sin\theta_{k}}\cos\gamma_{k}\\ \vdots \\ -e^{-j\left(M-1\right)\frac{\pi }{2}\sin\theta_{k}}\cos\gamma_{k} \end{array}\right]$$(25)$$a_{y_{k}}=\left[\begin{array}{c} q_{k}\cos\theta_{k}\sin\gamma_{k}e^{j\eta_{k}}\\ e^{-j\frac{\pi }{2}\sin\theta_{k}}q_{k}\cos\theta_{k}\sin\gamma_{k}e^{j\eta_{k}}\\ \vdots \\ e^{-j\left(M-1\right)\frac{\pi }{2}\sin\theta_{k}}q_{k}\cos\theta_{k}\sin\gamma_{k}e^{j\eta_{k}} \end{array}\right]$$

According to Equations (Equation 24) and (Equation 25), we can obtain the expression(26)$$a_{x_{k}}\mu_{k}=a_{y_{k}}$$
where $\mu_{k}=-q_{k}\cos\theta_{k}\tan\gamma_{k}e^{j\eta_{k}}$; then, there is(27)$$a_{x_{k}}\mu_{k}=a_{y_{k}}$$
where $A_{x}=\left[a_{x_{1}},a_{x_{2}},\cdots ,a_{x_{K}}\right]$, $A_{y}=\left[a_{y_{1}},a_{y_{2}},\cdots ,a_{y_{K}}\right]$, $\Omega =diag\{u_{1},u_{2},\cdots ,u_{k}\}$. The signal subspaces $E_{\mathrm{sx}}$ and $E_{\mathrm{sy}}$ correspond to subarray 3 and subarray 4, which consist of rows 1,3,⋯,2M−1 of $E_{s}$ and rows 2,4,⋯,2M of $E_{s}$, respectively, noting that(28)$$E_{\mathrm{sx}}=C_{3}A_{x}F$$(29)$$E_{\mathrm{sy}}=C_{4}A_{y}F=C_{3}A_{x}\Omega F$$
where F is a full-rank K×K-dimensional transformation matrix. From Equations (Equation 28) and (Equation 29), we obtain(30)$$E_{\mathrm{sy}}=E_{\mathrm{sx}}F^{-1}\Omega F$$(31)$$E_{\mathrm{sx}}^{+}E_{\mathrm{sy}}=F^{-1}\Omega F$$

Let $\Lambda =E_{\mathrm{sx}}^{+}E_{\mathrm{sy}}$; then, we have(32)$$\Lambda =F^{-1}\Omega F$$

Λ and Ω are similar matrices with the same eigenvalues. The eigenvalue decomposition of Λ obtains the values on the diagonal of Ω. Since the eigenvector of each signal is unique with a mode of 1, this property is utilized to perform parameter matching. Specifically, the eigenvector obtained from DOA estimation is calculated, and the eigenvector from polarization estimation is used as an inner product to complete the matching process. The following Equations (Equation 33) and (Equation 34) can obtain the estimations of ${\hat{\gamma }}_{k}$ and ${\hat{\eta }}_{k}$:(33)$${\hat{\gamma }}_{k}=\arctan\left(\frac{abs[{\hat{u}}_{k}]}{-{\hat{q}}_{k}\cos{\hat{\theta }}_{k}}\right)$$(34)$${\hat{\eta }}_{k}=angle\left[{\hat{u}}_{k}{\hat{q}}_{k}^{-1}\right]$$
where ${\hat{u}}_{k}$ denotes the *k*th eigenvalue of Λ and abs[·] denotes taking the modulus value of the complex number. At this point, we can summarize the specific steps of the proposed algorithm as follows (Algorithm 1).
**Algorithm 1** Subarray Selection-Based Decoupled ESPRIT**Step1:** Use the received signal $\tilde{X}\left(t\right)$ to obtain an estimation of its covariance matrix $\hat{R}$.**Step2:** Apply eigenvalue decomposition to $\hat{R}$ and obtain the signal subspace $E_{s}$.**Step3:** Divide the array into subarray 1 and 2, then estimate the matrix Ψ to obtain ${\hat{\theta }}_{k}$ by Equation (Equation 19).**Step4:** Re-divide the array into subarray 3 and 4, then estimate the matrix Λ to obtain ${\hat{\mu }}_{k}=-{\hat{q}}_{k}\cos{\hat{\theta }}_{k}\tan{\hat{\gamma }}_{k}e^{j{\hat{\eta }}_{k}}$ by Equation (Equation 30).**Step5:** ${\hat{\theta }}_{k}$ and $\mu_{k}$ are parametrically matched and finally the estimations of ${\hat{\gamma }}_{k}$ and ${\hat{\eta }}_{k}$ are obtained by using Equations (Equation 33) and (Equation 34).

## 4. Simulation Results and Discussion

This section presents three experiments to evaluate the effectiveness and performance of the proposed method. The root mean square error (RMSE) is employed to assess the accuracy of the algorithm’s parameter estimation as follows:(35)$$RMSE=\sqrt{\frac{1}{KL}\sum_{k=1}^{K}\sum_{l=1}^{L}{\left({\hat{\chi }}_{k,l}-\chi_{k}\right)}^{2}}$$
where *L* stands for the count of Monte Carlo experimental simulations, ${\hat{\chi }}_{k,n}$ denotes the estimated DOA or polarization values of the *k*th signal in the *k*th Monte Carlo simulation experiment, and $\chi_{k}$ denotes the real DOA or polarization values of the *k*th signal.

The simulation study assumes that a PSA consists of M=12 groups of elements, with each group comprising two mutually orthogonal and non-collocated elements, arranged in a ULA configuration. The distance between neighboring elements is set to $d=\frac{\lambda }{4}$. The vector of mutual coupling coefficients between the elements is set to $c=\left[1,\;0.512+0.241i,\;0.234+0.176i\right]$ in all simulations.

### 4.1. Algorithm Effectiveness

In the first experiment, we verify the effectiveness of the proposed algorithm and compare its performance with the conventional undecoupled ESPRIT algorithm and reduced-dimension MUSIC (RD-MUSIC) algorithm for the estimation of the angle θ. A far-field narrowband signal is assumed to be incident on the array, with the signal parameters set as θ=46.50°, γ=47.37°, and η=57.48°. The number of snapshots is N=300, and the signal-to-noise ratio (SNR) is 10 dB. The effectiveness of the three algorithms in estimating the parameter θ is illustrated in Figure 7.

Figure 7 illustrates the estimation performance of the three algorithms for the angle θ. The black solid line represents the true value of θ, while the three color markers indicate the estimated values across multiple Monte Carlo simulations. The proposed method exhibits highly stable estimates that fluctuate only slightly around the true value, demonstrating minimal deviation and strong robustness against both noise and mutual coupling effects. In contrast, the ESPRIT algorithm suffers from significant estimation errors, with values noticeably deviating from the true line, indicating poor accuracy and a lack of robustness to these interferences. The RD-MUSIC algorithm performs better than ESPRIT but still exhibits a larger deviation from the true value compared to the proposed method, showing reduced estimation accuracy due to its sensitivity to mutual coupling.

These results confirm that the proposed method achieves the highest accuracy and stability among the three approaches, as its estimates consistently cluster around the true value. In contrast, ESPRIT suffers from severe deviations, making it highly susceptible to noise and mutual coupling effects, while RD-MUSIC, despite improvements over ESPRIT, still demonstrates greater estimation errors than the proposed method.

### 4.2. Effect of Mutual Coupling Coefficients on Estimation

This section analyzes the effect of the mutual coupling coefficients’ magnitude on parameter estimation and compares it with the conventional undecoupled ESPRIT algorithm and RD-MUSIC algorithm. Assume one far-field narrowband signal incident on the array with DOA θ=46.41° and polarization parameters γ=47.48°, η=57.54°. The SNR is fixed at 10 dB and the snapshot number N=300. Varying the magnitude of the mutual coupling coefficients $c_{2}=0.512+241i$ and $c_{3}=0.234+0.176i$ from 0 to 1, 1000 Monte Carlo experiments are conducted to compare and observe the RMSE. The comparison of the RMSE of the mutual coupling coefficients magnitude for the theta estimation of the three methods is shown in Figure 8, and the comparison of the RMSE of the magnitude of the mutual coupling coefficients for the gamma estimation and the eta estimation is shown in Figure 9.

Figure 8 illustrates the variation in the RMSE of DOA parameter θ estimation across different algorithms as the magnitude of the mutual coupling coefficient increases from 0 to 1. The proposed algorithm maintains consistently low RMSE values throughout the entire range of mutual coupling coefficients. In contrast, the RMSE of the conventional undecoupled ESPRIT algorithm increases rapidly with the mutual coupling coefficient. This demonstrates its inability to handle high mutual coupling scenarios effectively, resulting in inaccurate parameter estimation. Meanwhile, the RD-MUSIC algorithm consistently shows lower estimation accuracy compared to the proposed method. Note that when the mutual coupling coefficient is very small, the undecoupled algorithm slightly outperforms the proposed method. This occurs because the undecoupled algorithm utilizes more array elements and, in low-coupling scenarios, the element count has a larger influence on estimation accuracy. However, the proposed algorithm achieves effective decoupling processing by sacrificing a portion of the array elements. As the mutual coupling coefficients increase, it substantially outperforms the undecoupled algorithm, maintaining lower estimation error and demonstrating superior stability and robustness.

In Figure 9, the estimated RMSEs of the three algorithms for the polarization angles γ and η under different mutual coupling coefficients are shown, respectively. The estimation errors of the traditional un-decoupled algorithm and RD-MUSIC algorithm gradually increase with the increase in mutual coupling coefficients. In contrast, the algorithm of this paper maintains a lower error under the wide range of mutual coupling conditions. This result further verifies that the proposed algorithm can achieve more stable parameter estimation in an environment with significant mutual coupling effects, and still possesses significant anti-interference performance even when the array element’s number is reduced.

### 4.3. Effect of Signal-to-Noise Ratio on Estimation

In the last section, we investigate the effect of the magnitude of the SNR on the parameter estimation and compare it with the conventional undecoupled ESPRIT algorithm and RD-MUSIC algorithm. It has the same parameters of the incident signal as for the last experiment. The snapshot number N=300 and SNR from 0 dB to 20 dB were used for 1000 Monte Carlo experiments to compare and observe the RMSE. The comparison of the RMSE of the magnitude of the SNR for the theta estimation of the three methods is shown in Figure 10, and the comparison of the RMSE of the magnitude of the SNR for the gamma estimation and the eta estimation is shown in Figure 11.

It can be observed from Figure 10 that the estimation errors of all algorithms decrease as the SNR increases. However, at a lower SNR, the estimation error of the traditional undecoupled ESPRIT algorithm is much larger than that of the decoupled algorithm proposed. Although the RD-MUSIC algorithm demonstrates stability, its estimation accuracy is still inferior to the proposed algorithm. In contrast, the proposed algorithm can significantly reduce the estimation error in the presence of mutual coupling and exhibits high robustness in low-SNR conditions.

Figure 11 further demonstrates the RMSE results of the three algorithms in estimating the auxiliary polarization angles γ and η. With the increase in SNR, the errors of this paper’s algorithm in γ and η estimation are significantly lower than those of the undecoupled algorithm and RD-MUSIC algorithm, and the advantages are more obvious, especially at a low SNR. This result demonstrates that the proposed algorithm is superior in effectively suppressing the influence of the mutual coupling effect on parameter estimation, and can maintain high estimation accuracy and stability under various SNR conditions.

### 4.4. Discussion

The experimental results confirm that the proposed method outperforms ESPRIT and RD-MUSIC in terms of estimation accuracy, robustness to mutual coupling, and noise resilience. Table 1 summarizes the key performance metrics, including RMSE ranges under different mutual coupling and SNR conditions. These results demonstrate that the proposed method effectively mitigates mutual coupling effects, maintains low estimation errors under varying SNRs, and is a more reliable solution for practical DOA and polarization estimation.

## 5. Conclusions

This paper presents a new method for multidimensional parameter estimation in scenarios with mutual coupling interference based on a non-collocated crossed dipole array model. By integrating the ESPRIT method with a subarray selection strategy, the algorithm effectively eliminates the mutual coupling matrix, achieving accurate parameter estimation while reducing the impact of mutual coupling. The simulation results demonstrate that the proposed method performs robustly under various SNR and mutual coupling conditions. Compared to the proposed method, the estimation error of the ESPRIT algorithm is higher by approximately −7.9~454.28% when the mutual coupling magnitude increases from low to high levels, and by 148.55~2329.22% under varying SNR conditions. Similarly, the RD-MUSIC algorithm exhibits higher estimation errors by 29.91~83.22% during mutual coupling magnitude variations and by 33.73~351.11% under SNR changes. The proposed method shows strong potential for practical applications, especially in environments where mutual coupling effects are unavoidable. Future work may focus on further optimizing the algorithm efficiency and exploring its application to larger and more complex array configurations.

## Figures and Tables

**Figure 1 sensors-25-01391-f001:**
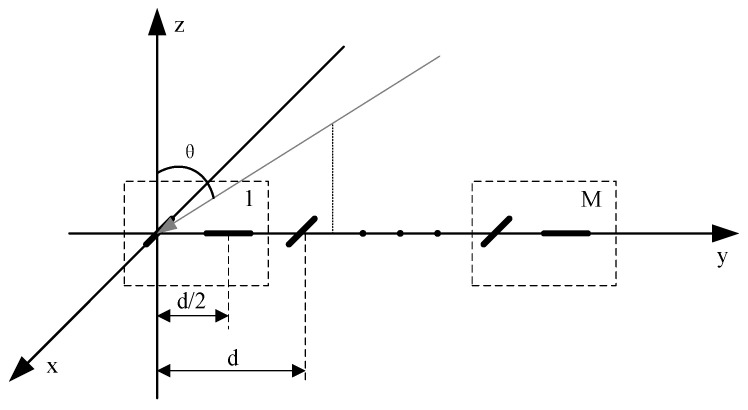
Polarization-sensitive array model.

**Figure 2 sensors-25-01391-f002:**
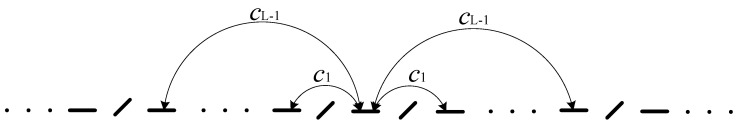
Influence of element mutual coupling parallel to the *x*-axis.

**Figure 3 sensors-25-01391-f003:**
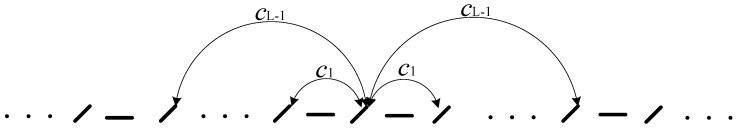
Influence of element mutual coupling parallel to the *y*-axis.

**Figure 4 sensors-25-01391-f004:**
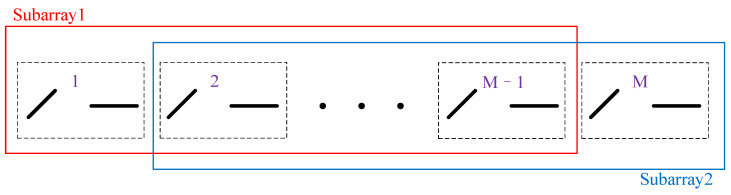
Regular subarray selection.

**Figure 5 sensors-25-01391-f005:**
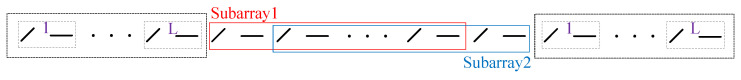
Subarray selection for estimating DOAs.

**Figure 6 sensors-25-01391-f006:**
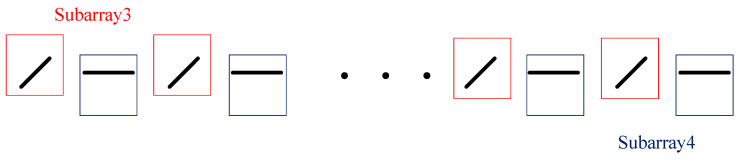
Subarray selection for estimating polarization parameters.

**Figure 7 sensors-25-01391-f007:**
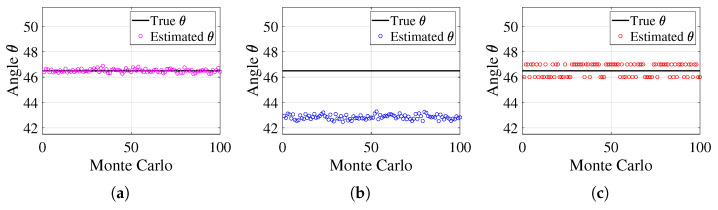
Comparison of three methods. (**a**) Proposed method. (**b**) ESPRIT. (**c**) RD-MUSIC.

**Figure 8 sensors-25-01391-f008:**
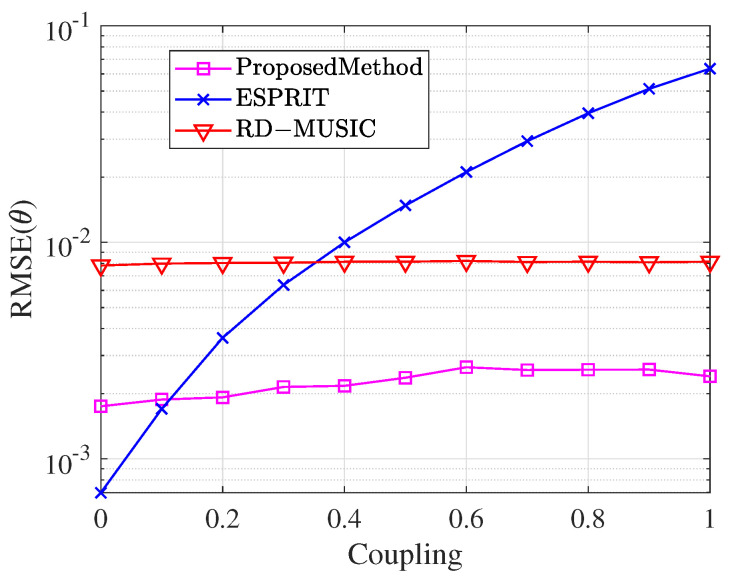
Comparison of RMSE for θ while coupling magnitude changes.

**Figure 9 sensors-25-01391-f009:**
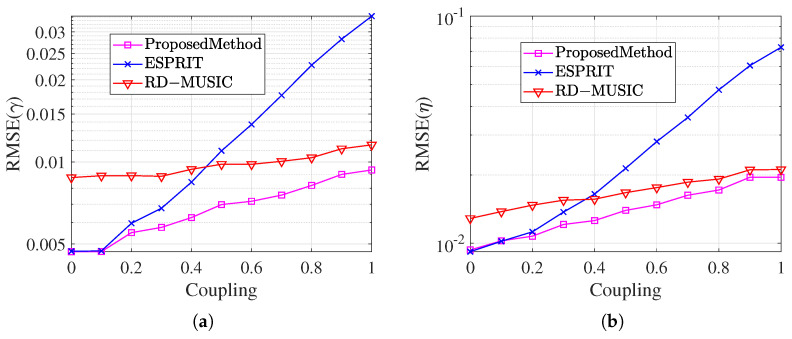
Comparison of RMSE for γ and η while coupling magnitude changes. (**a**) RMSE result for angle γ. (**b**) RMSE result for angle η.

**Figure 10 sensors-25-01391-f010:**
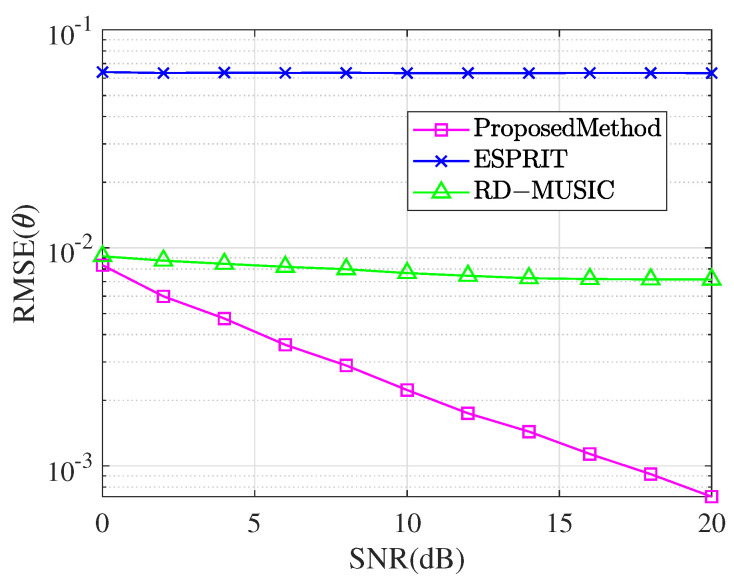
Comparison of RMSE for θ while SNR changes.

**Figure 11 sensors-25-01391-f011:**
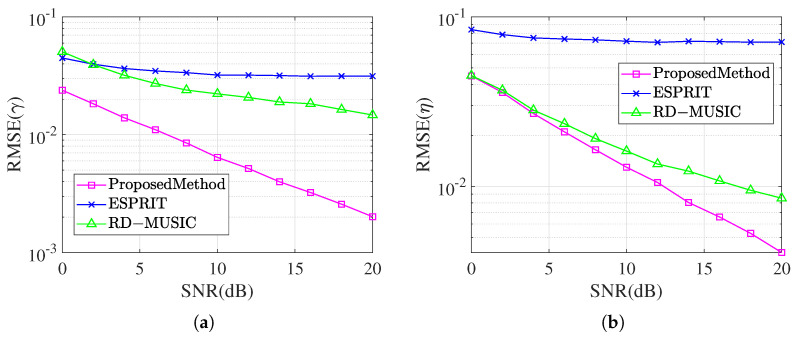
Comparison of RMSE for γ and η while SNR changes. (**a**) RMSE result for angle γ. (**b**) RMSE result for angle η.

**Table 1 sensors-25-01391-t001:** Comparison of different estimation algorithms.

Algorithm	Effectiveness	RMSE (rad) vs. Mutual Coupling	RMSE (rad) vs. SNR
Proposed Method	High	0.0055~0.0102	0.0259~0.0023
ESPRIT	Failed	0.0051~0.0566	0.0644~0.0552
RD-MUSIC	Moderate	0.0101~0.0133	0.0346~0.0103

## Data Availability

Data are contained within the article.

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
