# Peer review of "Robust Direction of Arrival and Polarization Parameter Estimation in Mutual Coupling Scenario with Non-Collocated Crossed Dipole Arrays"

_sensors, 2025, doi:10.3390/s25051391_

Round 1

Reviewer 1 Report

Comments and Suggestions for Authors

This paper introduces a novel low-complexity sparse reconstruction algorithm for joint estimation of DOA and polarization parameters in polarization-sensitive arrays, and provides complexity analysis and simulation experiments that validate the proposed algorithm's ability to balance complexity and accuracy effectively. Overall, the paper demonstrates significant advantages in complexity and offers valuable research contributions. However, in my reading, I found the following points that could be improved or need further clarification:

1. In the paper, there is inconsistency in the use of singular and plural forms of polarization parameters. For example, the abstract uses the plural form "polarization parameters," while the Section V uses the singular form "polarization parameter".

2. In equation (3), the meanings of the symbols ⊗ and âŠ™ are not clearly defined. It is recommended to ensure that each symbol is properly defined to avoid confusing the readers.

3. The paper does not demonstrate significant advantages in the performance of polarization parameters estimation. Please clarify the limitations of this method or suggest potential improvements.

Reviewer 2 Report

Comments and Suggestions for Authors

The authors propose a scheme for DoA and polarization estimation in a scenario where mutual coupling is present. The main contribution is in the proposal of a modified ESPRIT algorithm seeking for low complexity, and which is based on subarray selection for the mutual coupling compensation. The manuscript tackles an interesting topic, but it should be improved before I can consider it for publication. Here I summarize the main points that require attention.

l.1 and on: Every time you use an acronym, you should always declare it first.

l.15-19: Such claims should be supported by references from the literature.

l.21-22: The authors can include a very brief revision of DoA estimation algorithms and systems. Along with subspace separation algorithms, I suggest also including systems that perform the DoA estimation in hardware with lightweight approaches based on phase interferometry. For instance, you can refer to systems for Angle of Arrival based on full-hardware I/Q approach.

l.39-58: When performing such literature revision, the proposed techniques should be compared by briefly highlighting the advantages and drawbacks of each approach.

l.61-62 and on: Please perform a careful revision of the English language in the text to improve the understanding.

l.87: The authors can include in their literature revision also approaches that aim to compensate for both mutual coupling and circuit realization artifacts in DoA/AoA estimation applications, e.g. linear techniques for artifacts correction and compensation in angle of arrival estimation.

l.252: The simulation section should be revised and improved since lots of information that directly impacts the claim of the article is either absent or poorly motivated. For instance, details such as the RF frequency, the sampling rate, etc. are totally missing. Moreover, for the mutual coupling estimation and compensation, it is necessary to make some hypotheses on the type of antenna employed, since it strongly impacts the mutual coupling entity. Also, the authors should clearly state the simulation environment and setup they are using, to make the result reproducible. Last, as for the results discussion section, the authors should include a summary table of the performance achieved, and make some comparison with other approaches in the literature.

l.336: A summary of the performance quantifying the viability of the proposed approach should be provided in the conclusions.

Comments on the Quality of English Language

Please perform a careful revision of the English language in the text to improve understanding.

Round 2

Reviewer 2 Report

Comments and Suggestions for Authors

The authors addressed my previous concerns. Still, there are some minor aspects the authors should take into account before I can recommend the paper for publication. First of all, I suggest the authors perform a new and careful English re-check since there are some phrases whose construction can be improved for clarity. Also, In some Figures (e.g. Figure 7) the authors can use larger fonts to improve readability. Table 1 can be improved by including some results which are described in the conclusions. In fact, the evaluation in Table 1 is too qualitative. Some values may help in understanding the actual improvements. Moreover, I suggest the authors rely mostly on references from international and recognized conferences and journals in order to rely on a more robust bibliography.

Comments on the Quality of English Language

I suggest the authors perform a new and careful English re-check since there are some phrases whose construction can be improved for clarity.
